# The Role of Perioperative Interleukin-6 Serum Levels on Liver Dysfunction and Infectious Complications After Hepatectomy—A Systematic Review

**DOI:** 10.3390/cancers17193120

**Published:** 2025-09-25

**Authors:** Alexander Kofler, Marlene Trattner, Vivien Mairinger, Iveta Urban, Kjetil Søreide, Stefan Stättner, Florian Primavesi

**Affiliations:** 1Department of Visceral, Transplant and Thoracic Surgery, Center for Operative Medicine, Medical University of Innsbruck, 6020 Innsbruck, Austria; a.kofler@i-med.ac.at; 2Department of General, Visceral and Vascular Surgery, Salzkammergutklinikum Vöcklabruck, 4840 Vöcklabruck, Austria; marlene.trattner@hotmail.com (M.T.); mairingervivien@gmail.com (V.M.); 3Department of Surgery, Ordensklinikum Linz, 4010 Linz, Austria; ivetaurban@yahoo.com; 4Department of Gastrointestinal Surgery, Stavanger University Hospital, 4068 Stavanger, Norway; ksoreide@icloud.com; 5Department of General and Visceral Surgery, Hepatobiliary Unit, Kepler University Hospital GmbH, Johannes Kepler University Linz, 4021 Linz, Austria; stefan.staettner@kepleruniklinikum.at

**Keywords:** liver dysfunction, post-hepatectomy liver failure, interleukin-6, complications, infection

## Abstract

Liver resection is a key treatment for many malignant and benign diseases but carries the risk of post-hepatectomy liver failure and infectious complications, both of which significantly affect outcomes. Reliable early markers to identify patients at risk remain limited. Interleukin-6 is a cytokine that rises after surgery in response to both tissue regeneration and infection, making its interpretation clinically challenging. This systematic review analyzed available studies on perioperative serum interleukin-6 levels and their relationship to complications. Findings suggest that interleukin-6, measured particularly on the first postoperative day, may provide early warning of adverse outcomes. However, differences in study design, small sample sizes, and lack of validated cut-offs currently limit its routine clinical use. Further research is required to define standardized thresholds, optimal timepoints, IL-6 assays, and multi-marker strategies to improve postoperative monitoring and patient safety.

## Key Points

Serum interleukin-6 (IL-6) is used as a clinical marker for infection and inflammation. It also increases significantly after liver resection as a reflection of hepatic regeneration stimulus. Therefore, IL-6 may be used perioperatively as an indicator of imminent infection or liver dysfunction.This review assessed 12 studies (*n* = 589 patients) on perioperative serum IL-6 levels and their association with pre-defined post-hepatectomy liver failure (PHLF) or infection.Especially on the first postoperative day, IL-6 is significantly associated with subsequent complications, but cut-offs are not well-defined. At present, a single-marker, single-timepoint assessment does not seem sufficient to discriminate between infection and impaired regeneration.Heterogeneous cohorts and clinical endpoints, as well as insufficient consideration of confounding factors, warrant adequately powered, well-designed future studies.

## 1. Introduction

Interleukin-6 (IL-6) plays an important role in both the acute-phase response against infections as well as liver regeneration (Figure 1). In clinical settings, postoperative serum IL-6 along other inflammatory markers may be used to monitor patients early after liver resection. This could help to possibly detect imminent postoperative infections, but interpreting IL-6 dynamics can be challenging due to a concurrent peak in the immediate liver regeneration phase.

As a pro-inflammatory cytokine IL-6 is produced at local tissue sites by various cells [1,2]. Under inflammatory conditions, IL-6 induces acute phase protein transcription pathways (C-reactive Protein/CRP, fibrinogen, etc.) in the liver (Figure 1A) [3]. Moreover, IL-6 is a major factor in hepatic regeneration [2]. Animal studies showed impaired hepatocyte proliferation and regeneration with subsequent liver failure after partial hepatectomy in IL-6 -/- knock out mice. Liver proliferation in these mice can be restored by injecting recombinant IL-6 preoperatively, indicating the high potential of IL-6 in liver regeneration [4]. In this regenerative context (Figure 1B), a significant increase in tumor-necrosis factor alpha (TNF-α) activates various transcription factors in Kupffer cells and hepatocytes, including nuclear factor kappa-light-chain-enhancer of activated B cells (NF-κB) and signal transducer and activator of transcription 3 (STAT3) [5]. Consequently, IL-6 is produced in high amounts in Kupffer cells, with a consecutive increased release into surrounding tissues. Specifically, hepatocytes then respond by initiating growth and promoting liver regeneration. The acute phase response in the liver usually does not exceed 48 h, after which the organism returns to normal hepatic function, except in chronic states of inflammation [2].

Post-hepatectomy liver failure (PHLF) remains a challenging complication in patients undergoing liver resection and represents the major cause of mortality [5,6,7]. Compared to treatment options in other types of end-stage organ failure such as dialysis in kidney disease, to date, aside from salvage liver transplantation, no causal effective therapy exists for advanced liver dysfunction [8]. Therefore, early detection of patients at risk for liver dysfunction is critical in order to immediately apply preventive or supportive treatment [5,7]. Amongst several definitions for PHLF the International Study Group of Liver Surgery (ISGLS) criteria are most accepted (see Refs. [6,9,10]) based on bilirubin and prothrombin time outside normal ranges on postoperative day 5 or later. More recently, markers such as serum lactate, CRP, and antithrombin III were demonstrated to identify imminent PHLF as early as within 24 h of surgery [11,12,13]. This is crucial as PHLF may lead to subsequent complications such as infection, bleeding, and kidney failure [10,13,14]. Intriguingly, insufficient postoperative liver regeneration not only occurs due to primarily limited volume (small-for-size) of the future liver remnant (FLR), but can also be caused secondarily by infection, especially abdominal sepsis [5,15]. As infectious complications (bile leakage, wound infections, pneumonia) are common after major hepatectomies, serum marker profiles to differentiate between early sepsis and imminent PHLF would be clinically helpful [11,16].

The aim of this systematic review therefore was to evaluate the value of IL-6 in identifying a possible complicated perioperative course in liver resection patients, with a specific focus on PHLF and infection.

## 2. Methods

### 2.1. Search Strategy and Data Generation

The protocol for the study has not been registered. A systematic literature search between January 2000 and June 2025 was executed to investigate the correlation between perioperative IL-6 levels, post-operative liver dysfunction, and infection. The search was performed in accordance with the Preferred Reporting Items for Systematic Reviews and Meta Analyses (PRISMA) guidelines, using the PubMed, Embase, and Cochrane databases [17]. The MeSH terms (Medical Subject Headings) used in a multi-field search are given in Appendix A. Only human studies reporting original data written in the English-language with available full-text manuscripts were included; congress abstracts were excluded.

The primary outcome of this review was association of perioperative IL-6 levels and liver dysfunction, preferable according to clearly pre-defined criteria. The secondary outcomes included correlation of IL-6 with infectious complications, overall morbidity, and mortality. Furthermore, studies were screened for (patient- and disease-related or surgical) co-factors that could potentially confound measured levels of perioperative serum IL-6.

### 2.2. Data Extraction and Quality Assessment

Retrieved publications were pooled into the citation manager Endnote (Clarivate Analytics, London, UK); duplicates were identified and removed. Abstracts were screened and double-checked by two investigators to identify relevant studies. The resulting full-text papers were analyzed, and the reference list of selected studies was screened for additional publications fulfilling the inclusion criteria. Consent was achieved through discussion about remaining studies.

Data extraction from the defined studies included: study design, country of origin, cohort size and configuration, operative procedure type (open or laparoscopic), histopathological diagnosis, timepoints of IL-6 measurement, main outcomes measured, overall morbidity and mortality rates, PHLF assessment and definition, infectious complication assessment, and follow-up duration. Median IL-6 levels in different cohorts and at different timepoints were evaluated, as well as the predictive potential for PHLF or infectious complications. All randomized-controlled trials (RCTs) were assessed and graded for methodological quality according the Jadad Oxford scale [18], and all non-RCT studies according the Newcastle–Ottawa scale [19].

## 3. Results

### 3.1. Study Selection and Characteristics

Appendix A shows the PRISMA literature-screening diagram. A total of 1289 records were found during study identification with multi-field search. After applying exclusion criteria and removal of duplicates, a final number of 12 publications with a collective 589 patients (range 14–128) were included in this review (Table 1) [20,21,22,23,24,25,26,27,28,29,30,31]. These studies (all published between 2001 and 2023) were all prospective single-center observational cohort studies, except for five (partly secondary analysis of) RCTs with comparably low numbers of patients and potential statistical limitations [24,25,26,27,28].

Concerning the timepoints of IL-6 measurement (Figure 2), ten studies measured IL-6 pre-operatively. On the day of surgery, four studies additionally assessed IL-6 immediately before resection and eight in the first 12 h afterwards. Eleven studies measured IL-6 on postoperative day (POD)1, and nine studies included measurements beyond POD1. Most (*n* = 10) publications evaluated postoperative overall morbidity or mortality. Six of the studies specifically reported (liver) organ dysfunction rates, [20,22,23,24,27], although only two in particular evaluated PHLF as defined by the ISGLS classification, [30,31] and three studies had applied their own individual definition [20,22,23].

### 3.2. Methodological Quality

The methodological quality assessment is summarized in Appendix A. All included studies were rated Oxford level of evidence 2b. The methodological quality of the included RCTs was low (maximum 2 points of the 5-point Jadad scale), except for the study from Schwarz et al. from 2015, which reached 3 points (intermediate quality) due to an appropriate randomization process and description of the withdrawals [24].

All non-RCTs reported baseline patient characteristics and seven of them specified the enrollment timeframe. Indication for liver surgery was given in all studies, although details on pre-existing liver disease and intraoperative blood loss and operation time were often not presented. All but two studies assessed outcomes within the timeframe of hospital stay. None of the studies fulfilled all features of the Newcastle–Ottawa scale, indicating a limited study quality and a high risk of bias. This was mainly due to inappropriate assessment of whether the factors of interest were present preoperatively, inadequate demonstration of true comparability of the control groups, lack of adjustment for potential confounding factors, or insufficient adequacy of follow-up.

Appendix A summarizes information on the interleukin-6 assay types applied in the individual publications included in this review. Most studies used commercially available ELISA (enzyme-linked immunosorbent essay) kits of different manufacturers. However, some authors reported the use of other methods such as cytometric bead arrays or (electro)chemiluminescence enzyme immunoassays.

### 3.3. Perioperative Factors Associated with IL-6 Serum Levels

Table 2 summarizes studies included in the review with overall outcomes and IL-6 values stratified by different subgroups and timepoints. Seven studies reported on perioperative factors potentially confounding IL-6 serum levels in univariable analysis [20,22,23,24,25,26]. Preoperative jaundice was associated with significantly higher IL-6 levels in the first hours after resection [20,23] and on POD1 [22], as were intraoperative blood transfusions [22]. Also, longer surgeries and major hepatectomies resulted in higher IL-6 values immediately after the end of the operation [22,23], while in one study, there was no apparent correlation of extend of resection with postoperative IL-6 levels [31].

#### 3.3.1. Impact of Surgical Techniques

While application of inflow-control (Pringle maneuver) did not influence IL-6 levels [23], one RCT showed that the type of parenchymal transection (cavitron ultrasonic surgical aspirator (CUSA) vs. stapler resection) did affect intraoperative portal-vein and hepatic-vein IL-6 levels, but not intraoperative or postoperative systemic IL-6 values [24].

Another RCT by Li et al. from 2015 (*n* = 26) [25] showed that a laparoscopic compared to an open surgical approach significantly lowers the postoperative increase in peripheral blood IL-6 serum levels at end of the operation as well as on POD1 (IL-6: open 116.19 pg/mL vs. laparoscopic 73.60 pg/mL; *p* < 0.05). As a limitation, the frequency and type of complications per group were not described in this study.

Similarly, a RCT published by Kasai et al. 2018 (*n* = 40) compared the impact of laparoscopic vs. open procedures on inflammatory response and angiogenesis in patients undergoing hepatectomy for colorectal liver metastases (CRLM) [29]. Both groups had a significant postoperative increase in IL-6 levels in relation to preoperative baseline values, as previously demonstrated by Li et al. Again, laparoscopic resection showed lower, however not significant, IL-6 levels compared to open resection on POD1 (39.1 vs. 98.9 pg/mL; *p* = 0.358).

In one recent study by Ammann et al., patients (*n* = 46) underwent liver resection also including partial hepatectomy after the application of future remnant liver augmentation strategies. Resections were stratified by resection extent (major or minor per Brisbane 2000 criteria), with no difference in postoperative day 5 IL-6 levels between the two groups, indicating that the resection volume does not necessarily lead to sustained IL-6 levels after the immediate postoperative period [31].

#### 3.3.2. Impact of Preexisting Liver Disease

In a small, prospective case–control study (*n* = 14) from 2003 [21], the acute phase response and cytokine levels in patients undergoing liver resection for hepatocellular carcinoma (HCC) due to chronic hepatitis B were compared to those of healthy donors for living liver transplantation. Postoperative laboratory results showed a significant rise in IL-6 levels on POD1 in both groups, with a lower increase in the healthy donor group (IL-6 preop/POD1/POD2: 4.12/47.51/25.81 pg/mL) compared to HCC patients (11.50/141.90/109.50; all *p* < 0.05). In contrast, perioperative CRP levels were not significantly different between the two groups. The authors concluded that healthy liver donor cases have more capacity to adapt to surgical stress than patients with underlying liver disease.

### 3.4. Primary Endpoint: IL-6 Dynamics, PHLF, and Infection

One of the first studies suggesting a correlation between circulating cytokines, organ dysfunction, and infections in patients undergoing open liver resection was published by Kimura et al. in 2006 [22]. Of 128 patients enrolled, 86 had non-infectious complications (bleeding, pneumothorax, bowel paralysis), 31 had postoperative infection, and 11 had infection-related single or multiple organ failure. IL-6 was significantly higher on POD1 in cases with subsequent postoperative infections (including intraabdominal abscess, pneumonia, wound infection, bacteremia, and cholangitis) or any organ dysfunction (respiratory, liver, renal, cardiovascular, hematologic, neurological). Particularly, POD1 IL-6 levels were 401 pg/mL in patients with no infections/organ dysfunction versus 1027 pg/mL in those with organ dysfunction (*p* < 0.01) and 598pg/mL with infectious complications (*p* < 0.05). Also, a correlation was drawn between the duration of surgery and IL-6 increase on the day of the operation (POD0). No specific analysis between IL-6 levels and predefined PHLF was performed in this publication. However, multivariable logistic regression suggested that the IL-6 level on POD1 was a significant, independently associated variable for incident postoperative infections, as well as any organ dysfunction. A calculated cut-off value of >702 pg/mL on POD1 was linked to the occurrence of subsequent infections with an odds ratio (OR) of 3.65 (95% confidence interval/95%-CI: 1.47–9.07; *p* = 0.005).

In 2011, Strey et al. assessed perioperative cytokine levels in a series of 26 liver resection patients performed in a German University Hospital [23]. The study comprised a heterogeneous cohort of minor and major hepatectomies for a variety of different entities. The authors showed that larger and more time-consuming liver resections lead to pronounced increases in postoperative IL-6 levels. Also, patients with various post-operative complications (bile leak, wound or abdominal infections, organ failure) had higher median levels of IL-6 at all explored timepoints. In terms of specific liver function assessment, they did not apply pre-specified liver dysfunction criteria such as ISGLS-PHLF. However, the study demonstrated that impairment of coagulation function (PT cut-off applied: <56%) or bilirubin clearance (maximum bilirubin postoperative >2 mg/dL) was associated with significantly increased IL-6 levels within the first 24 h postoperatively. As a limitation, these findings were not further evaluated for potential interactions with preexisting confounders by multivariable analysis or case matching.

In a small RCT, Sarin et al. 2016 (*n* = 20) studied the impact of a three-day short course of preoperative oral statins on liver reperfusion injury in major hepatectomy, as this has been shown to be beneficial in previous animal models [26]. They assessed transaminases (AST/ALT) and cytokines (IL-1, IL-6, CRP, and TNF alpha). POD1 IL-6 levels were significantly lower in the statin intervention group compared to the control group (83.83 vs. 144.43 pg/mL; *p* = 0.001). A potential association of IL-6 levels and pre-defined liver dysfunction was not assessed in the study, nor were infectious or overall complications separately studied.

In 2020, a prospective cohort study by Arisaka et al. (*n* = 68 major hepatectomies) specifically focused on combining plasma markers for prediction of PHLF as defined by the ISGLS classification [30]. Several perioperative liver function-associated parameters, transaminases, and inflammatory markers were explored via the Youden’s index analysis to assess sensitivity and specificity for liver dysfunction prediction. Nine patients (13.3%) experienced clinically relevant (CR-) PHLF (ISGLS grade B/C) and were compared to all cases without CR-PHLF. While preoperative serum IL-6 levels were comparable in both groups, significant differences were noted postoperatively, with markedly higher levels in patients with liver failure on POD1, 3, and 5, with a peak on POD1. The area-under-the-curve (AUC) for POD1-IL-6 for CR-PHLF was 0.791 (compared to 0.929 for POD1-Bilirubin), with a calculated cut-off of 106.5 pg/mL (77.8% sensitivity and 91.5% specificity). The combination of IL-6 with preoperative platelet count further increased its’ predictive potential (AUC = 0.838). Furthermore, CR-PHLF in this study had a clear association with severe morbidity (Clavien–Dindo grade ≥ 3) and mortality rate (33%).

Most recently, a 2023 prospective observational study by Ammann et al. (*n* = 46 minor and major liver resections) showed that IL-6 levels rose early after both minor and major resections, but differences were driven by postoperative complications rather than resection extent. In patients without PHLF, IL-6 declined over time, whereas it remained elevated in PHLF, resulting in a marked POD5 difference (*p* < 0.001) and higher perioperative AUC (*p* = 0.005). After excluding cases with PHLF or major morbidity, no IL-6 differences between minor and major resections were observed, indicating that impaired liver regeneration accounts for sustained IL-6 elevation [31].

### 3.5. Secondary Outcomes: IL-6 Dynamics and Overall Postoperative Morbidity

A secondary subgroup analysis of a RCT published by Cata et al. in 2017 (*n* = 41) reported a significant increase in IL-6 levels on POD 1, 3, and 5 after surgery compared to preoperative baseline measurements [28]. However, there was no significant difference in median cytokine levels on POD1, 3, and 5 in patients with (*n* = 14/34%) or without complications. However, the authors included a broad variety of complications (neurological, cardiac, respiratory, renal, etc.) and neither specifically assessed PHLF or infectious complications separately, nor did they differentiate between mild and severe complications.

Another RCT comprising *n* = 40 patients conducted by Schwarz et al. from 2015 performed a comparison between two parenchymal transection methods (CUSA vs. stapler) in terms of perioperative inflammatory markers and transection speed [24]. Prolonged operation time correlated with higher IL-6 levels, with a peak on POD1. Additionally, intraoperative samples were drawn from the hepatic and portal vein to evaluate differences depending on the sample location. No significant difference in serum IL-6 assessed in peripheral blood was seen between the two transection technics. However, the authors showed a significant difference related to transection type (CUSA vs. stapler) regarding levels from the portal vein (40.6 vs. 14.4 pg/mL; *p* = 0.026) and hepatic vein (29 vs. 13.2 pg/mL; *p* = 0.042). No specific assessment of IL-6 and postoperative PHLF-rates, infectious complications, or overall morbidity was performed.

Another cohort study [24] conducted by Schwarz et al. in 2017 (*n* = 40) investigated the link between inflammatory response and oxidative stress during liver resection as well as complications [27]. They discovered that patients experiencing severe complications had significantly higher IL-6 levels on POD3 compared to patients without severe morbidity. Again, no specific evaluation of a link between IL-6 levels and PHLF or infectious complications was reported.

In their study from 2001, Das et al. evaluated 100 consecutive patients undergoing hepatectomy, stratified into three groups of pre-existing liver parenchymal quality (preoperative obstructive jaundice, liver cirrhosis, and normal liver) [20]. They showed that cirrhotic and jaundiced liver patients had significantly higher median serum IL-6 levels at 12 h postoperatively (267 pg/mL and 641 pg/mL) compared to normal liver parenchyma cases (179 pg/mL). However, these results were not further analyzed for interaction potential with other influential factors such as necessity of major resections, blood loss, etc., though, within the groups of liver cirrhosis and obstructive jaundice patients, those developing any complication had significantly higher IL-6 values compared to cases without complications. Although the overall rate of PHLF and infectious complications in the total cohort was reported, no specific analysis in terms of association with IL-6 levels was performed.

## 4. Discussion

Perioperative care of liver resection patients is complex. The alertness of clinicians to be able to anticipate imminent complications early and initiate adequate diagnostic and therapeutic actions are key to achieve favorable outcomes [32,33,34]. Especially in major hepatectomies, the postoperative liver function mainly determines the overall course and subsequent morbidity [7,13,35]. Avoidance of infections is crucial to ensure an uncomplicated regeneration of the FLR and the patient [15]. Therefore, markers related to postoperative liver parenchyma growth and infection such as IL-6 are increasingly assessed in translational research.

IL-6 is a driver for producing acute-phase proteins such as CRP, hepcidin, or fibrinogen in the liver, and is produced by immune cells, hepatocytes, and Kupffer cells. In the perioperative phase of liver resection, IL-6 is a major factor in hemostasis, angiogenesis, and regeneration. It is involved in thrombopoietin homeostasis and platelet count restoration pathways as well as angiotensinogen gene expression during liver regeneration [36,37]. Therefore, it may be pertinent to draw a clinical correlation between liver function, complications, and systemic IL-6 levels. Overproduction of IL-6 potentially indicates an overwhelming systemic inflammatory response syndrome (SIRS), resulting in organ failure, an impaired immune system, and consecutive severe infections [22].

This systematic review revealed a limited number of publications specifically addressing the link between IL-6, PHLF, and infectious complications. While postoperative CRP has recently been assessed in a large multicenter project with appropriate methodology [11], no robust evidence so far has been published on how IL-6 serum measurements can differentiate between PHLF and infection early after liver resection. Specifically, only a few studies have included multivariable analysis to account for potential confounders such as underlying liver disease, comorbidities, indication, extent, and technical aspects of resection, or other factors that could influence perioperative IL-6 levels per se. With commonly less than 50 patients included per study, most publications are underpowered to assess CR-PHLF with event rates of 5–15% according to the literature [5,7,13,38,39]. Therefore, to date, IL-6 should not be routinely used in isolation to differentiate between infection and PHLF. Its greatest current value lies in signaling patients at risk for adverse events, highlighting the need for confirmatory prospective trials.

The majority of studies furthermore utilized heterogeneous cohorts, assessed different outcome parameters with varying study endpoints and limited follow up periods, and did not apply established PHLF criteria. This precluded a meta-analysis, specifically in relation to a specific cut-off for early postoperative IL-6 levels to predict ISGLS-CR-PHLF in pre-defined cohorts (e.g., minor versus major resections) [31,40].

Examining IL-6 serum marker dynamics, several studies confirmed significantly increased levels at various postoperative timepoints compared to baseline. Specifically, IL-6 on POD1 seems clinically useful to aid in decision-making. While a calculated cut-off value of >702 pg/mL on POD1 was linked to four-times-increased rates of subsequent infections in a consecutive, heterogeneous cohort by Kimura et al. [22] a cut-off of 106 pg/mL (77.8% sensitivity and 91.5% specificity) was reported in major hepatectomy patients by Arisaka et al. to predict ISGLS-PHLF with an AUC of 0.791 [30]. These large differences in cut-off estimation for different purposes (infection vs. PHLF) demonstrate the limited applicability of single markers such as IL-6 in not clearly defined clinical scenarios. Most certainly instead, multi-biomarker patterns would be more appropriate, as has been suggested by Arisaka et al., who analyzed a combination of POD1-IL-6 and preoperative thrombocytes for PHLF prediction [30]. For example, it could be particularly interesting to assess the ratio of CRP- or leukocytes-to-IL-6.

A number of studies included in this review suggest that IL-6 levels are influenced by several perioperative factors, including the surgical approach (open versus minimally invasive), the extent/volume of resection, occurrence of ischemia and reperfusion injury, intraoperative blood loss, and the baseline hepatic function. Although acknowledged through univariable analysis, few studies adjusted for these variables in multivariable models, limiting the ability to isolate IL-6’s independent effect. Future studies should be designed with pre-specified multivariable models to account for these confounders.

Moreover, there was marked variability in the timing of IL-6 measurement across studies, with sampling ranging from immediate postoperative measurements to POD14. This heterogeneity in sample timepoints, together with variable IL-6 assay tools used, complicates direct comparison and may contribute to inconsistent findings. Future prospective, multicenter studies should adopt not only pre-defined primary endpoints and PHLF definitions (ideally as proposed by the ISGLS [6]), but also harmonized IL-6 measurement schedules and uniform assay platforms across different centers to facilitate valid pooled analyses. Accordingly, optimal timepoints and reliable cytokine assays would need to be defined upfront before study initiation.

Aiming for a more granular and dynamic liver regeneration marker assessment over several days, Ammann et al. demonstrated that IL-6 levels rise immediately after surgery in both minor and major resections but only decline in patients with adequate liver regeneration. In PHLF, IL-6 remained persistently elevated, with a marked difference evident by POD5, underscoring its role as an early inflammatory marker of impaired recovery. Notably, IL-6 closely paralleled the postoperative increase in glucagon-like peptide-2 (GLP-2), while GLP-1 exhibited an inverse pattern. These hormone dynamics, more pronounced in PHLF, were independent of resection extent or complication severity, indicating a connection between metabolic and regenerative capacity rather than surgical trauma. The alignment of sustained IL-6 elevation with GLP-2 rise suggests a mechanistic link between inflammatory signaling and regeneration in the context of lipid-metabolism. Together, IL-6 and GLP-2 trajectories may provide complementary biomarkers for early identification and postoperative monitoring of patients at risk for PHLF [31]. As a consequence, serial, perioperative measurements of IL6 and other regeneration markers implementing machine-learning algorithms might further help to detect characteristic patterns of regeneration versus infection in the future [41].

Prevention, or at least early detection and mitigation, of the vicious circle of imminent SIRS, infection, and PHLF is a major measure to increase safety of liver resections and improve patient outcomes. Glucocorticoid administration represents an option to regulate inflammatory response and hereby potentially influence perioperative liver function and regeneration [42,43]. A recent meta-analysis of RCTs assessing perioperative glucocorticoids in patients undergoing liver surgery revealed a lower risk of overall complications (RR 0.77, 95% CI 0.64–0.92) as well as improvement in early postoperative (POD1-3) laboratory markers associated with inflammation (IL-6/CRP) and liver function (international normalized ratio/total bilirubin/albumin) [42]. Pooled analysis of IL-6 levels showed significantly decreased levels in the glucocorticoid administration cohorts, with a mean difference of −53 pg/mL on POD1 and −22 pg/mL on POD3 (both *p* < 0.001). These results underline that perioperative steroid administration can attenuate excessive inflammatory responses, inhibit hepatic ischemia and reperfusion injury, and might potentially decrease the rate of PHLF. To date, the heterogeneity of study designs and glucocorticoid doses applied with moderate overall evidence still result in a weak grade of recommendation in the most recent ERAS^®^ (enhanced recovery after surgery) guidelines [44].

In summary, preoperative IL-6 serum levels were associated with known baseline risk factors for poor outcomes after liver resection. At several perioperative timepoints, IL-6 values are linked to overall morbidity, PHLF, and infection-related complications, although an independent link has not yet been demonstrated and validated cut-offs are missing. Comparable patterns of IL-6 dynamics beside further cytokines have recently been reported in other contexts [45,46,47]. Overall, evidence remains exploratory, and conclusions should be interpreted with caution due to small sample sizes and risk of bias and be considered hypothesis-generating rather than definitive. At present, IL-6 measuring should remain a research tool rather than part of routine postoperative monitoring until validated in large, multicenter trials. Additional research is required to understand the clinical role of serum IL-6 dynamics in combination with other inflammatory and liver function biomarkers to identify clinical useful patterns in different liver surgery scenarios, ideally through machine learning algorithms.

The existing clinical evidence is almost entirely correlative, with perioperative IL-6 levels measured in serum but rarely linked to mechanistic tissue-level validation. Future translational studies could address this gap by combining clinical monitoring with perioperative liver biopsies to assess molecular markers of IL-6 activity, such as STAT3 phosphorylation, SOCS3 expression, or microRNA levels [48]. This would help to further analyze the regulatory complexity of the IL-6 signaling pathway, aiming to clarify whether IL-6 elevations in individual patients primarily reflect regenerative signaling, systemic inflammation, or both. Ultimately, this information would serve as an important basis for future study protocols targeting cytokine, inflammation, or liver regeneration intervention measures.

## 5. Conclusions

Perioperative interleukin-6 levels are consistently associated with postoperative complications after liver resection, with values on the first postoperative day showing the greatest predictive potential. However, current evidence is limited by small sample sizes, heterogeneous cohorts, and lack of validated cut-offs. Interleukin-6 should therefore be regarded as an exploratory marker. Future multicenter studies with standardized protocols and integration into multimarker panels are required to define its role in predicting post-hepatectomy liver failure and infection.

## Figures and Tables

**Figure 1 cancers-17-03120-f001:**
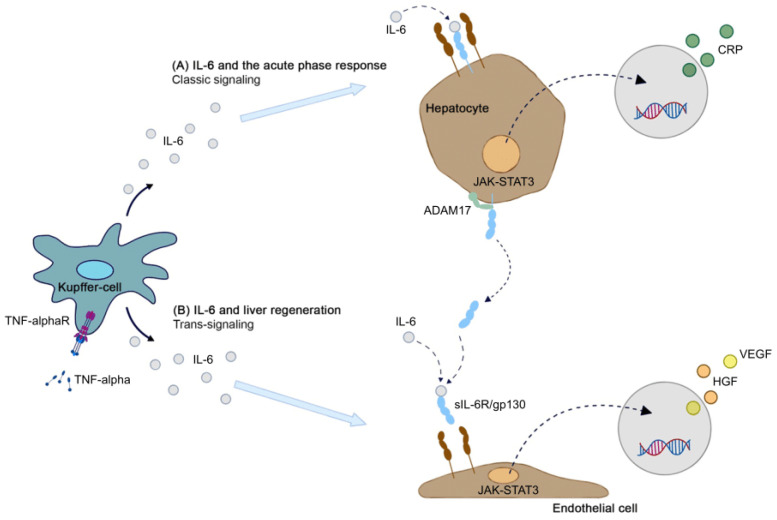
IL-6 signaling pathway in the (**A**) acute phase response and in (**B**) liver regeneration. ADAM17 = ADAM metallopeptidase domain 17; CRP = C-reactive protein; EC = endothelial cell; gp130 = glycoprotein 130; HGF = hepatocyte growth factor; IL-6 = interleukin-6; JAK-STAT = Janus kinase–signal transducer and activator of transcription protein; sIL-6R = soluble IL-6 receptor; TNF-alpha = tumor necrosis factors alpha; TNF-alphaR = TNF-alpha receptor; VEGF = vascular endothelial growth factor.

**Figure 2 cancers-17-03120-f002:**
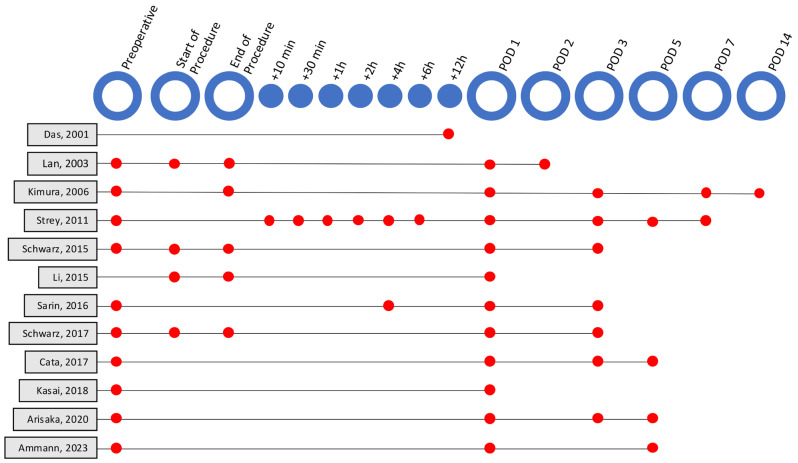
Timeline showing different perioperative timepoints of IL-6 measurements in studies included in this review. Studies are listed according first author and year of publication. POD = postoperative day [20,21,22,23,24,25,26,27,28,29,30,31].

**Table 1 cancers-17-03120-t001:** Study characteristics of the 12 publications finally included in the review. CCC = cholangiocellular carcinoma; CCSC = circulating cancer stem cells; CRLM = colorectal liver metastases; EOP = end of procedure; GBC = gallbladder cancer; GLP = glucagon-like peptide; HCC = hepatocellular carcinoma; ICU = intensive care unit; IL = interleukin; ISGLS = international study group for liver surgery; LDLT = living donor liver transplantation; LM = liver metastases not specified in detail; NELM = neuroendocrine liver metastasis; NNECR = non-neuroendocrine non-colorectal liver metastasis; POD = postoperative day; post-OP = postoperative; Pre-OP = preoperatively before the day of surgery; SOP = start of procedure; TNF-α = tumor-necrosis-factor alpha; * postoperative hyperbilirubinema > 10 mg/dL, termed liver failure in presence of hepatic encephalopathy, ascites, and PTT < 40%; ^#^ serum bilirubin > 240 µmol/l; ^§^ PT < 56% or maximum bilirubin postoperative > 2 mg/dL. ** Overall complications were assessed according the Claviend–Dindo classification, no specific PHLF criteria defined.

Study	Type	Country	*n*	Surgical Approach	Histopathological Diagnosis	Time of IL 6 Measurement	Main Outcomes Measured	PHLF Assessed? (Specific Criteria)	Infectious Complications Assessed?	Follow Up Duration
Das et al. (2001) [20]	Prospective cohort	Japan	100	Open	HCC, CCC, GBC, CRLM, NCRLM, benign	POD 0: EOP + 12 h	Morbidity, mortality	Yes (individual) *	Yes	In-hospital stay
Lan et al. (2003) [21]	Prospective cohort	Taiwan	14	Open	LDLT, HCC	PreOP; POD 0: SOP, EOP; POD 1,2	Morbidity, mortality, intraoperative complications	No	No	In-hospital stay
Kimura et al. (2006) [22]	Prospective cohort	Japan	128	Open	HCC, CCC, CRLM, NCRLM	Pre-OP; POD 0: EOP; POD 1,3,7,14	Morbidity, mortality, organ dysfunction, infections, cytokine levels	Yes (individual) ^#^	Yes	In-hospital stay
Strey et al. (2011) [23]	Prospective cohort	Germany	26	Open	HCC, CCC, CRLM, benign	Pre-OP; POD 0: EOP + 10 min-6 h (6×); POD 1,3,5,7	Morbidity, length of stay, liver function	Yes (individual) ^§^	Yes	In-hospital stay
Schwarz et al. (2015) [24]	RCT	Austria	40	Open	HCC, CRLM, Echinococcus	Pre-OP; POD 0: SOP; EOP; POD 1,3	Morbidity, mortality, length of stay, length of ICU, operative time	Yes (no specific criteria) **	Yes	30 days
Li et al. (2015) [25]	RCT	China	26	Open and laparoscopic	HCC	POD 0: SOP, EOP; POD 1	Peak IL-6, IL-8 and TNF-α; CCSC counts	No	No	In-hospital stay
Sarin et al. (2016) [26]	RCT	India	20	Open	HCC, GBC, CCC, benign	Pre-OP; POD 0: EOP + 4 h, POD 1,3	Ischemia reperfusion injury via cytokine levels in patients with or without statins	No	Yes	In-hospital stay
Schwarz et al. (2017) [27]	Prospective cohort	Austria	40	Open	HCC, LM, Echinococcus	Pre-OP; POD 0: SOP, EOP; POD 1,3	Oxidative stress and inflammation, morbidity and mortality	Yes (no specific criteria) **	Yes	30 days
Cata et al. (2017) [28]	RCT	USA	41	Open	CCC, GBC, CRLM NCRLM	Pre-OP; POD 1,3,5	Morbidity	No	Yes	In-hospital stay
Kasai et al. (2018) [29]	RCT	Belgium	40	Open and laparoscopic	CRLM	Pre-OP; POD 1	Morbidity, length of stay, overall survival	No	Yes	In-hospital stay
Arisaka et al. (2020) [30]	Prospective cohort	Japan	68	Open	CCC, HCC, LDLT, LM	Pre-OP; POD 1,3,5	Morbidity, mortality, length of stay, liver function	Yes (ISGLS)	Yes	In-hospital stay
Ammann et al. (2023) [31]	Prospective cohort	Austria/USA	46	Open	HCC, CCC, NELM, NNECR, benign	Pre-OP; POD 1,5	GLP1 and GLP2 dynamics, PHLF, IL-6 dynamics, DPP4, lipid metabolism	Yes (ISGLS)	No	In-hospital stay

**Table 2 cancers-17-03120-t002:** Study outcomes, IL-6 values for different subgroups and timepoints, association with morbidity, and predictive potential for PHLF and infectious complications. AUC = area under the curve; CCC = cholangiocellular carcinoma; CRLM = colorectal liver metastases; CR-PHLF = clinically relevant post-hepatectomy liver failure (ISGLS Grade B/C); EOP = end of procedure; Hepatitis B-HCC = hepatitis B associated hepatocellular carcinoma patients; IC: infectious complications; ISGLS = international study group of liver surgery; LC = liver cirrhosis; n/a = not assessed; NIC: non-infectious complications; NL = normal liver; OD = organ dysfunction; OR = odd’s ratio; PHLF = post-hepatectomy liver failure; postop. = postoperative; PT = prothrombin time; * = *p* < 0.05, ** = *p* < 0.01, *** = *p* < 0.001, ^ns^ = non-significant. ^#^ approximate values are derived from depicted graphs as no specific numbers were reported in the publication text.

Publication	Overall Morbidity	Overall Mortality	Configuration Studied Groups	Median IL-6 Levels Studied Groups (pg/mL)	Median IL-6 Level w/wo Complications (pg/mL)	Regression/ROC Analyses Prediction of Event
Das et al. (2001) [20]	14%	4%	Normal liver (NL) vs. liver cirrhosis (LC) vs. obstructive jaundice (OJ)	12 h postop: 180 (NL) vs. 268 (CL) vs. 641 (OJ) **	Cirrhosis (12 h postop): 555/211 * Jaundice (12 h postop): 925/456 *	n/a
Lan et al. (2003) [21]	14.3%	0%	Hepatitis B-HCC vs. LDLT patients	POD0: 11.5 vs. 4.1 * EOP: 59.3 vs. 22.8 * POD1: 141.9 vs. 47.5 * POD2: 109.5 vs. 25.8 *	n/a	n/a
Kimura et al. (2006) [22]	49%	2%	Non-infectious complications (NIC) vs. infectious complications (IC) vs. organ dysfunction (OD)	NIC vs. IC. Vs. OD: POD0 EOP: 15.6 vs. 11.2 vs. 38.2 * POD1: 8.6 vs. 7.4 vs. 41 ** POD3: 11.2 vs. 7.5 vs. 3.5	n/a	Multivariable analysis: Postop. infection: POD1 OR: 2.43 ** Postop. OD: POD1 OR: 2.66 **
Strey et al. (2011) [23]	53.8%	n/a	Postop PT < 56% vs. >56% Postop bilirubin < 2 vs. >2 mg/dL	POD1: 420 vs. 110 *^#^ POD1: 400 vs. 120 *^#^	Any vs. no complication: POD1: 400 vs. 130 ^ns#^	n/a
Schwarz et al. (2015) [24]	37.5%	2.5%	CUSA vs. stapler hepatectomy	Intraoperative: 25.5 vs. 14.5 ^ns^	n/a	n/a
Li et al. (2015) [25]	n/a	n/a	Laparoscopic vs. open HCC resection	EOP: 73.6 vs. 116.2 * POD1: 121.8 vs. 192.4 *	n/a	n/a
Sarin et al. (2016) [26]	n/a	n/a	Atorvastatin group vs. Placebo	EOP + 4 h: 124.4 vs. 214.3 ** POD1: 83.8 vs. 144.4 ** POD3: 28.82 vs. 54.02 **	n/a	n/a
Schwarz et al. (2017) [27]	37.5%	2.5%	No/minor vs. severe complications	n/a	POD3: 24.6 vs. 53.7 **	n/a
Cata et al. (2017) [28]	34.1%	0%	41 consecutive resections for CCC and CRLM. No control group	POD0: 28 vs. POD1: 216 ** POD3: 128 ** POD5: 70 **	POD1: 254 vs. 206 ^ns^ POD2: 98 vs. 128 ^ns^ POD3: 76 vs. 59 ^ns^	n/a
Kasai et al. (2018) [29]	25%	0%	Open vs. laparoscopic hepatectomy	POD1 98.9 vs. 39.1 ^ns^	n/a	n/a
Arisaka et al. (2020) [30]	28%	4%	ISGLS PHLF B/C (13.2%) vs. no CR-PHLF	POD0: 20 vs. 10 ^ns#^ POD1: 210 vs. 50 *^#^ POD3: 150 vs. 40 *^#^ POD5: 80 vs. 30 *^#^	n/a	CR-PHLF: Cut-off POD1: 106.5 AUC: 0.791 (95% CI 0.562–1.000)
Ammann et al. (2023) [31]	26%	6.5%	No PHLF vs. PHLF, minor vs. major resection,	No PHLF vs. PHLF PRE: 5.0 vs. 10.8 ^ns^ POD1: 170 vs. 320 ^ns#^ POD5: 45 vs. 270 ***^#^	n/a	n/a

## Data Availability

All data used for this systematic review is available in the manuscript, tables, and Appendix A.

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
