# Peer review of "The Role of Perioperative Interleukin-6 Serum Levels on Liver Dysfunction and Infectious Complications After Hepatectomy—A Systematic Review"

_cancers, 2025, doi:10.3390/cancers17193120_

Round 1
Reviewer 1 Report
Comments and Suggestions for Authors
In the manuscript by Kofler et al., titled “The role of perioperative interleukin-6 serum levels on liver dysfunction and infectious complications after hepatectomy a systematic review,” The study is informative and contributes valuable insights into an important clinical issue. I have several specific comments intended to help the authors further enhance the clarity, rigor, and overall impact of the manuscript.
- Does IL-6 useful in differentiating between post-hepatectomy liver failure (PHLF) and infectious complications in early post-operative period?
- how stable are the IL-6 time points of measurement across research, and what implications does this variability have on the comparison and validity of the combined outcomes?
- Do the studies included in it have a high-quality data (sample size, study and bias control) to draw conclusion about IL-6 as predictive markers for PHLF or infection?
- Does the existence of perioperative confounders (e.g. surgical technique, underlying liver disease, intraoperative blood loss) have significant effects on IL-6 levels and has this been sufficiently considered in the review?
- Is there enough evidence to support integrating of IL-measurement into routine post operative monitoring protocols following liver resection?
- Could the authors stratify IL-6 dynamics more precisely surgical subgroups (e.g., minor vs. major resections, pre-existing liver disease?
- Do authors propose a standardized universal time and a universal method for perioperative IL-6 measurement in future studies to allow comparability?
- Are there specific examples of how machine learning methods are supposed to be configured (what types of data, inputs, etc.) so that they could recognize clinically significant patterns in IL-6?
Author Response
In the manuscript by Kofler et al., titled “The role of perioperative interleukin-6 serum levels on liver dysfunction and infectious complications after hepatectomy a systematic review,” The study is informative and contributes valuable insights into an important clinical issue. I have several specific comments intended to help the authors further enhance the clarity, rigor, and overall impact of the manuscript.
1. Does IL-6 useful in differentiating between post-hepatectomy liver failure (PHLF) and infectious complications in early post-operative period?
We acknowledge that the ability of IL-6 to discriminate between PHLF and infection remains uncertain. Our revised discussion now clearly states that current evidence supports IL-6 as a possible marker of postoperative inflammation and regeneration but does not yet provide robust data on the discriminatory power to early predict imminent occurrence of PHLF or infection-related complications. We emphasize that this is a key gap to be addressed by future prospective studies (see line 43 onwards in abstract, line 62 onwards in key points, line 518 onwards in discussion).
2. how stable are the IL-6 time points of measurement across research, and what implications does this variability have on the comparison and validity of the combined outcomes?
We agree that variability in timing is a critical limitation. In the revised discussion section, we explicitly note the wide heterogeneity of sampling schedules and explain how this variability complicates direct comparison and synthesis of findings. We also recommend that future studies standardize measurement time points and also assay platforms to enable data analysis (line 446 onwards)
3. Do the studies included in it have a high-quality data (sample size, study and bias control) to draw conclusion about IL-6 as predictive markers for PHLF or infection?
We appreciate this observation. Our quality assessment (Supplementary Table S1) indicates that most studies are small, single-center, and carry a notable risk of bias. We have revised the discussion section to explicitly state that the current evidence base is exploratory and that conclusions regarding predictive utility must be considered hypothesis-generating (lines 518 onwards).
4. Does the existence of perioperative confounders (e.g. surgical technique, underlying liver disease, intraoperative blood loss) have significant effects on IL-6 levels and has this been sufficiently considered in the review?
Thank you for raising this point. We already discussed confounders in section 3.3 and mentioned heterogeneity of studies regarding multivariable analysis in discussion and also clarified that most included studies did not adjust for these confounders in multivariable models, limiting causal interpretation. Nevertheless, we fortified the call for multivariable adjustment in lines 439 onwards)
5. Is there enough evidence to support integrating of IL-measurement into routine post operative monitoring protocols following liver resection?
We agree that current evidence is insufficient to recommend routine IL-6 monitoring in clinical practice. We have revised the manuscript to state that IL-6 measurement should currently be considered to be further investigated for validation in large, prospective trials (abstract line 43 onwards, discussion line 415 onwards and line 518 onwards).
6. Could the authors stratify IL-6 dynamics more precisely surgical subgroups (e.g., minor vs. major resections, pre-existing liver disease?
We thank the reviewer for this suggestion. We addressed this stratification in table 2, in section 3.3.1 and 3.3.2. and also further pointed at this topic in the revised discussion in line 439 onwards.
7. Do authors propose a standardized universal time and a universal method for perioperative IL-6 measurement in future studies to allow comparability?
Not at present. We have specifically stated, that further comparative studies would be necessary before recommending certain time points or IL-6 assay methods (discussion line 446 onwards).
8. Are there specific examples of how machine learning methods are supposed to be configured (what types of data, inputs, etc.) so that they could recognize clinically significant patterns in IL-6?
We sincerely appreciate this thoughtful comment. Our team is currently conducting a study that leverages machine learning to analyze and integrate a wide range of variables related to IL-6. This approach aims to provide a more comprehensive understanding of the underlying patterns and relationships. Once the study is completed, we plan to share the results through an appropriate peer-reviewed publication.
Reviewer 2 Report
Comments and Suggestions for Authors
The review addresses a clinically relevant question—whether IL-6 can distinguish between infection and post-
hepatectomy liver failure (PHLF). The review advocates for combining IL-6 with other biomarkers to improve specificity
and notes the need for randomized trials. Future studies must standardize endpoints and adjust for confounders.
1. This study appears to have aimed to address the question of "The value of IL-6 serum levels to differentiate
infection from imminent post-hepatectomy liver failure (PHLF) remains unclear". However, the review's conclusion seems to deviate from the research objective.
2. Although "overall study quality was low, with high risk of bias", the authors conclude that IL-6 levels may detect imminent infectious complications or PHLF early after liver resection. It is not very convincing.
3. The review suggests POD1 IL-6 levels are promising, but ideal sampling times remain undefined.
4. The review is transparent about its limitations and sets a clear roadmap for future research. However, there are several issues regarding IL-6 measurement across studies, including the use of different commercial kits and varying time points.
5. Some of the references are too old.
Author Response
The review addresses a clinically relevant question—whether IL-6 can distinguish between infection and post-hepatectomy liver failure (PHLF). The review advocates for combining IL-6 with other biomarkers to improve specificity and notes the need for randomized trials. Future studies must standardize endpoints and adjust for confounders.
- This study appears to have aimed to address the question of "The value of IL-6 serum levels to differentiate infection from imminent post-hepatectomy liver failure (PHLF) remains unclear". However, the review's conclusion seems to deviate from the research objective.
We thank the reviewer for this observation. We have revised the conclusion section to better align with our stated objective, clarifying that while IL-6 shows potential as a biomarker, the current evidence is insufficient to definitively distinguish infection from PHLF. We now emphasize that the review’s primary finding is the identification of critical knowledge gaps and the proposal of sort of research roadmap rather than a definitive clinical recommendation. (abstract line 43 onwards and discussion line 518 onwards)
- Although "overall study quality was low, with high risk of bias", the authors conclude that IL-6 levels may detect imminent infectious complications or PHLF early after liver resection. It is not very convincing.
We agree and have softened our language to avoid overinterpretation. This change ensures that our conclusion reflects the high risk of bias we identified. (abstract results and conclusions, key points, discussion line 521 onwards)
- The review suggests POD1 IL-6 levels are promising, but ideal sampling times remain undefined.
We identified POD1 as one of the few corresponding time points across studies, as basically all of them included sampling on this day. We appreciate this important point and have expanded our discussion section to explicitly note that optimal sampling time points for IL-6 remain undefined. Also, we recommend that future studies systematically compare serial time points to identify the most informative measurement window. (lines 449 onwards in discussion)
- The review is transparent about its limitations and sets a clear roadmap for future research. However, there are several issues regarding IL-6 measurement across studies, including the use of different commercial kits and varying time points.
Thank you for this comment. We have analysed all studies in detail regarding the IL-6 assay applied (summarized in the new Supplementary Table S3), and have added a paragraph on this information in the results 3.2 section. (line 217 onwards) We further strengthened the discussion section by highlighting variability in IL-6 assays and timing as key contributors to heterogeneity. We also recommend that future research report assay type, lower limit of detection, and timing explicitly to allow for more meaningful comparisons. (lines 451 onwards)
- Some of the references are too old.
We thank the reviewer for this observation. Our reference list intentionally includes both relevant older studies that established the clinical importance of IL-6 in liver surgery and the most recent research available (including several from the past five years). We believe this combination provides both historical context and an up-to-date synthesis of the field
Reviewer 3 Report
Comments and Suggestions for Authors
This systematic review focuses on perioperative serum interleukin-6 (IL-6) levels, aiming to analyze their role in predicting post-hepatectomy liver failure (PHLF) and infectious complications. The authors included 12 studies (involving 589 patients) and found that IL-6 levels on postoperative day 1 show promise as an early prognostic marker for infectious complications. However, they also highlighted significant limitations in existing studies making it impossible to draw definite clinical recommendations.
Major:
- The Introduction and Discussion sections address IL-6’s dual role in liver regeneration and inflammation, but all cited studies are correlative. Instead of only focusing on "associations," the authors should supplement experimental validation using human tissue samples. For example, perform immunohistochemical staining and Western blot analysis of p-STAT3 on postoperative liver biopsy specimens from patients with and without PHLF.
- The review identifies "pre-existing liver disease" and "surgical trauma" as major confounders, yet it fails to include studies that adequately address these factors in multivariable models. Future studies must be designed with pre-specified multivariable analyses.
- A critical flaw in the current evidence base is the inability to conduct a meta-analysis due to high heterogeneity in study endpoints. Future studies should adopt internationally recognized criteria (e.g., the ISGLS PHLF criteria mentioned in PMID: 21236455) and pre-define primary endpoints.
- Supplementary Tables S1 and S2 confirm the authors’ claim of "low study quality," but this critical limitation is not fully utilized. Before attempting a meta-analysis of IL-6, two steps are necessary:Consider the cut-off values for IL-6, Conduct a formal assessment of bias risk.
- The authors propose that a "multi-marker approach" would be more effective, but this claim lacks robust validation. A mandatory experiment is needed: compare the predictive value of IL-6 alone versus a panel of markers]measured on postoperative day 1.
- Introduction or Discussion could be supplemented with:The signaling pathway activity of interleukin-6 (IL-6) depends not only on the activation of signal transducer and activator of transcription 3 (STAT3) induced by tumor necrosis factor alpha (TNFα), but also on the fine regulation of negative regulatory factors such as suppressor of cytokine signaling 3 (SOCS3). Exogenous interventions can affect this pathway by modulating the microRNA-221 (miR-221)/SOCS3 axis, thereby controlling inflammatory outcomes. This finding illustrates the "regulatory complexity of the IL-6 signaling pathway" and provides a reference for subsequent discussions on the mechanisms underlying perioperative dynamic changes in IL-6. Relevant literature, such as PMID: 35729047, can be cited.
Minor:
- Table 1 should add a column specifying the definition of PHLF used in each stud This is a key source of study heterogeneity.
- In the Discussion section, the authors state "no study has included multivariable analysis," which is overly absolute. This wording should be revised (e.g., to "few existing studies have incorporated multivariable analysis").
- In Table 1, the abbreviation "vLM" is not defined in the table’s legend, leaving its meaning unclear to readers. This definition must be added.
Author Response
This systematic review focuses on perioperative serum interleukin-6 (IL-6) levels, aiming to analyze their role in predicting post-hepatectomy liver failure (PHLF) and infectious complications. The authors included 12 studies (involving 589 patients) and found that IL-6 levels on postoperative day 1 show promise as an early prognostic marker for infectious complications. However, they also highlighted significant limitations in existing studies making it impossible to draw definite clinical recommendations.
Major:
1. The Introduction and Discussion sections address IL-6’s dual role in liver regeneration and inflammation, but all cited studies are correlative. Instead of only focusing on "associations," the authors should supplement experimental validation using human tissue samples. For example, perform immunohistochemical staining and Western blot analysis of p-STAT3 on postoperative liver biopsy specimens from patients with and without PHLF.
We acknowledge the importance of mechanistic validation of IL-6 signalling pathways in postoperative liver tissue. However, this work is a systematic review restricted to published clinical studies, and we did not perform laboratory experiments. We further clarified this limitation explicitly in the discussion (line 504 ff, also answering point 6 of your comments), highlighting that while correlative evidence dominates the literature, mechanistic validation remains an unmet need. We will also suggest that future translational studies using perioperative liver biopsies and molecular assays (e.g., STAT3 phosphorylation, SOCS3 regulation) are required to complement clinical observations.
2. The review identifies "pre-existing liver disease" and "surgical trauma" as major confounders, yet it fails to include studies that adequately address these factors in multivariable models. Future studies must be designed with pre-specified multivariable analyses.
Thank for this comment, which is absolutely correct. We have further emphasized this limitation in the discussion (line 417 ff).
3. A critical flaw in the current evidence base is the inability to conduct a meta-analysis due to high heterogeneity in study endpoints. Future studies should adopt internationally recognized criteria (e.g., the ISGLS PHLF criteria mentioned in PMID: 21236455) and pre-define primary endpoints.
We have pointed this out more specifically in the discussion (line 427 ff).
4. Supplementary Tables S1 and S2 confirm the authors’ claim of "low study quality," but this critical limitation is not fully utilized. Before attempting a meta-analysis of IL-6, two steps are necessary:Consider the cut-off values for IL-6, Conduct a formal assessment of bias risk.
As we did not have access to the underlying patient data for the individual studies and due to the reported heterogeneity we have not conducted a formal meta-analysis.
5. The authors propose that a "multi-marker approach" would be more effective, but this claim lacks robust validation. A mandatory experiment is needed: compare the predictive value of IL-6 alone versus a panel of markers]measured on postoperative day 1.
As our review is descriptive, we cannot provide experimental evidence. This would be indeed part of our next planned prospective, multicenter project - Compare individual markers and timepoints individually and then combine them with clinical variables through machine-learning approaches to predict PHLF.
6. Introduction or Discussion could be supplemented with:The signaling pathway activity of interleukin-6 (IL-6) depends not only on the activation of signal transducer and activator of transcription 3 (STAT3) induced by tumor necrosis factor alpha (TNFα), but also on the fine regulation of negative regulatory factors such as suppressor of cytokine signaling 3 (SOCS3). Exogenous interventions can affect this pathway by modulating the microRNA-221 (miR-221)/SOCS3 axis, thereby controlling inflammatory outcomes. This finding illustrates the "regulatory complexity of the IL-6 signaling pathway" and provides a reference for subsequent discussions on the mechanisms underlying perioperative dynamic changes in IL-6. Relevant literature, such as PMID: 35729047, can be cited.
Thank you for this point, we have added a paragraph accordingly and inserted the citation (See also suggestion Nr. 1 of your remarks).
Minor:
1. Table 1 should add a column specifying the definition of PHLF used in each stud This is a key source of study heterogeneity.
The column on PHLF assessed has been adopted to show specific criteria if available. This has also been further noted in the paragraph below the table.
2. In the Discussion section, the authors state "no study has included multivariable analysis," which is overly absolute. This wording should be revised (e.g., to "few existing studies have incorporated multivariable analysis").
Adopted.
3. In Table 1, the abbreviation "vLM" is not defined in the table’s legend, leaving its meaning unclear to readers. This definition must be added.
This was a typo, and it has been corrected.
Round 2
Reviewer 2 Report
Comments and Suggestions for Authors
I have no additional comments except to wonder why the changes in the revised version are at completely different locations (lines) than the ones indicated by the author in their response.
Reviewer 3 Report
Comments and Suggestions for Authors
Although the updates and content of the article are limited, the authors have sufficiently clarified the research limitations and responded to my comments point by point. I have no further suggestions.